# How Does Interactive Narrative Design Affect the Consumer Experience of Mobile Interactive Video Advertising?

Chao Gu [1], Shuyuan Lin [2], Wei Wei [3], Chun Yang [4], Jiangjie Chen [4], Wei Miao [3], Jie Sun [5] and Yingjie Zeng [6],*

[1] Academy of Arts & Design, Tsinghua University, Beijing 100084, China; cguamoy@my.honam.ac.kr
[2] Department of Media Design, Tatung University, Taipei 104, Taiwan; shuyuan@gm.ttu.edu.tw
[3] School of Textile Garment and Design, Changshu Institute of Technology, Changshu 215500, China; doublewei@cslg.edu.cn (W.W.); weimiao@cslg.edu.cn (W.M.)
[4] School of Design, Jiangnan University, Wuxi 214122, China; yc004009@gmail.com (C.Y.); chenjiangjie@jiangnan.edu.cn (J.C.)
[5] College of Arts and Design, Zhejiang A&F University, Hangzhou 311300, China; sunjie@zafu.edu.cn
[6] Department of Industrial Design, Honam University, Gwangju 62399, Republic of Korea
* Correspondence: zyjamoy@gmail.com

**Abstract:** With the rapid spread of mobile devices and the Internet, mobile interactive video advertising has become an increasingly popular means of accessing advertising information for a large number of users. Interactive narratives are advertisements that require collaboration between consumers and designers to complete the story. Interactive narratives influence marketing impact and the advertising experience. Building on previous research, this study delves deeper into the design methods of interactive narratives for mobile video advertisements. We developed various interactive narrative samples by controlling video quality parameters, content, and product involvement, and then measured consumer perceptions of these samples in a laboratory environment. The results indicate that six design methods for interactive narratives foster positive perceptions, immersion, and satisfaction in advertisements with low product involvement. For ads with a high degree of product involvement, two design methods can achieve positive consumer perceptions of interactive narratives. This study offers insights for businesses and interaction designers aiming to advance the commercial use of mobile interactive video advertising. At the same time, we propose a design method for mobile interactive video advertising that can also serve as an entry point for theoretical research on interactive narratives.

**Keywords:** interactive advertising; interactive narrative; digital marketing; consumer behavior



## 1. Introduction

### 1.1. Research Background

In the field of multimedia research, interactive narrative design is one of the most important directions [1]. With the rapid development of digital technology, storytellers have been given greater means and opportunities to present traditional linear narratives in innovative ways. While the design and marketing advantages of interactive videos over traditional linear videos remain unclear, it could be argued that interactive videos might influence consumers' preferences and purchasing intentions more due to co-creation and interactive experiences, while the artful sequencing by directors might be more impactful in linear videos. Nonetheless, as a novel marketing approach, interactive videos have seen enthusiastic experimentation from certain brands and designers. Notably, some studies suggest that interactive videos can increase user engagement by up to 591% and offer experiences that are 32% more memorable than linear videos, and have a conversion rate during viewing that is 11 times higher than traditional media [2]. When producing interactive videos, to achieve better design outcomes, interactive narratives stand out as

one of the pivotal factors worthy of exploration. Recently, interactive narratives have drawn the attention of researchers keen to explore their potential benefits. It is widely used in marketing, education, video games, contemporary theater, and the visual arts [3,4]. For instance, Dionisio et al. [5] developed an interactive narrative augmented reality picture book. This interactive picture book integrates features like scanning, gestures, and sounds, encouraging young readers toward environmental consciousness. Interactive narratives effectively convey cultural information and showcase tourist attractions [6]. During the digital viewing, visitors can engage with original commentaries presented as interactive narratives from tour guides. Feng et al. [7] emphasize the growing importance of examining interactive narrative structures as an increasing number of marketers adopt interactive videos for brand storytelling. An appropriate interactive narrative design method should take into consideration factors such as product characteristics, media types, and consumer preferences.

Interactive narratives come in various forms, ranging from simple branching structures to deeply immersive experiences. Various media showcase interactive narratives, such as interactive novels, movies, hypertext fiction, and narrative games [8]. In the field of marketing, mobile phone interactive video advertisements have emerged as a fresh medium for interactive narratives, aiming to enhance user experience and boost sales. Interactive narratives enable a two-way communication channel between enterprises and consumers. Enterprises employ interactive technology to move beyond the conventional one-way content delivery from businesses to consumers [9]. With interactive narratives, designers empower consumers to shape the story. Interactive experiences enrich consumer engagement beyond just the story's content [10]. Compared to designer-only content, co-created content offers more customization due to consumer participation. Mobile interactive video advertising has gained attention in the fields of management and consumer research based on these characteristics [11]. Many companies have widely adopted mobile interactive video ads to advertise and promote their products. However, there is currently no solid theoretical foundation guiding the creation of interactive narrative features, which would help designers develop these advertisements more efficiently.

According to Ryan [12], interactive narrative comprises text structure, interaction form, and the completeness of the story. The story's completeness can be simply classified as either 'yes' or 'no'. For marketing, mobile interactive video ads need to provide complete information as they are not just artistic endeavors [13]. Videos with complete stories are more shareable and viewable than those with incomplete narratives [14]. However, elements like text structure, interactive forms, and their combinations warrant closer scrutiny. Gu, Lin et al. (2022) emphasize the need for consumers to critically assess interactive narratives considering both text structure and interaction form. The text structure denotes how the story text is arranged, while the interaction form signifies the medium through which users input data and engage with the advertisement. Companies and researchers in the field of interaction design are slowly realizing the importance of systematic research on interaction narratives. Ferguson et al. [15] contend that both interaction and story structure are essential to virtual reality environments. The researchers noted that research into these two concepts was a vital part of developing excellent interactive works.

Mobile interactive video ads, characterized by interactive narratives, have shown significant potential in the marketing domain. For sustained positive marketing outcomes using interactive narratives, enterprises and designers urgently require rigorous theoretical support. However, subjectively assessing the effectiveness of ads that merge text structure with interactive elements poses challenges. Research shows that while highly immersive interactive storytelling enhances a viewer's overall positive sentiment, it might diminish their ability to recall specific story details [16]. The details that are overlooked in a story may be exactly what a business wishes to convey in its marketing efforts. This implies designers cannot merely opt for interactive narrative combinations based solely on heightened interactivity or improved storytelling. As a form of interactive media that combines both design and marketing implications, mobile interactive video advertising requires an

objective examination of the interactive narrative method. Primarily, this study delves into the two critical components of mobile interactive video ads: the text structure and interaction form. We strive to control variables to probe consumer perceptions and assess optimal combinations of interactive narratives. Our research emphasizes the synergy of text structure and interaction form, particularly within the enterprise and designer contexts. Additionally, we contribute to a deeper understanding of consumer behavior through interactive marketing.

### 1.2. Research Purpose

In our study, we crafted mobile interactive video advertisements featuring varied text structures and interaction forms, and evaluated consumers' perceptions regarding interactive narrative, subjective video quality assessment (SVQA), immersion, satisfaction, and purchase intention. Drawing from these indicators, we offer design recommendations for interactive narratives in mobile interactive video advertisements.

## 2. Literature Review
### 2.1. Product Involvement

Involvement is defined by the degree to which a specific situation engages an individual's interest and motivation [17]. For example, product, website, and social media. Our research specifically zeroes in on consumer product involvement. According to Das and Ramalingam [18], product involvement refers specifically to consumers' interest and enthusiasm when it comes to purchasing a particular product. A major component of explaining the concept of participation is the relevance of the consumer's internal needs and interests. According to Li et al. [19], high involvement signifies a strong connection between the product and the individual. In other words, the degree of product involvement not only depends on the attributes of the product, but also on consumer perceptions. It has been suggested by Ghali-Zinoubi and Toukabri [20] that consumers will be more involved with the product if it has irreplaceable value to them or if it impacts their lifestyle fundamentally. Consumers seem to make preliminary judgments on product involvement by integrating their needs, habits, and knowledge about the product. It has been argued in previous studies that product involvement can be defined as the relationship between consumers' perceptions of a product and their intrinsic needs, values, and interests. Peng et al. [21] demonstrate this in their study. Given this, product involvement can signify change. Individual needs and perceptions of products may differ between consumers in the same period of time. In different periods, product involvement may vary for the same consumer due to factors such as changes in living habits and cognitive functions. According to the Wu and Liang [22] study, consumers process information in different ways depending on product category and product involvement, thus influencing their purchasing decisions. For businesses and designers, factoring in product involvement is essential during the design and marketing promotion processes. As Habib et al. [23] found, consumers with high product involvement may consider shopping in a more thoughtful manner; however, consumers with low product involvement may make more impulsive purchases. In this case, designers must select appropriate design strategies based on consumer behavior patterns. In addition, relevant research on marketing has confirmed that product participation impacts consumer purchasing intentions [24]. Therefore, our research considers the possible impact of product involvement and includes it as one of the classification variables of advertising in order to make suggestions about the interaction design and marketing efforts of different products and enterprises at the same time.

### 2.2. Interactive Narrative

The distinction between interactive and linear narratives lies primarily in the user's engagement: in interactive narratives, users have agency within the digital system [25]. When compared with traditional narratives that focus more on the story itself, interactive narratives offer designers new possibilities when telling stories. The current interactive

technology supports a wide range of scenarios due to the development of interactive technology, making the user experience more comprehensive and realistic [26]. The surge in Internet usage and technological advancements in mobile devices have further propelled the popularity of interactive narratives. Increasingly, companies and designers are considering interactive narrative as a marketing and design strategy. In academia, researchers are also becoming increasingly interested in this kind of storytelling method that has gradually spread from the field of gamification to a broader range of application areas. As an example, Mercer et al. [27] introduced an interactive narrative system to bolster civic and moral education, allowing children to cultivate critical thinking via immersive play. As a result of this digital narrative, children are able to practice and develop their abilities to think and express themselves through playful experiences. De la Fuente Prieto et al. [28] discuss design approaches for future cross-media interactivity. According to them, the cognitive transformation of users in the virtual world has contributed to the development of a worldview, and interactive narratives have the potential to spread content across different media and construct virtual worlds. As Cardona-Rivera et al. [29] put forth, interactive narratives are dynamic: the story evolves through the interplay between the user and the system, culminating in a unique narrative molded by user decisions. In their view, interactive narratives carry an emotional power due to engagement, interruption, and inevitability. It is considered that text structure is one of the most important attributes to consider when designing an interactive narrative information system [30]. It is important to understand how the designer sets the text structure and how the user selects the interactive node. Both of these variables contribute to the plot's arrangement and occurrence. In other words, as the audience experiences the story, it is also editing it. Moreover, as stated by Ryan [12], the form in which users interact and the way in which they interact also affect the effectiveness of interactive narratives. An interactive form's design determines how the user will interact with the interactive narrative. An effective use of interactive storytelling can assist marketing efforts in driving innovative publicity and sales strategies. According to previous studies, interactive storytelling can stimulate the creativity and imagination of young people [31]. Particularly in video media, where technology and art are closely linked, viewers are able to participate in and contribute to the development of the story [32]. We examined consumers' perceptions of interactive narratives in mobile interactive video advertisements in order to find a reasonable method of designing an interactive narrative that meets both user needs and marketing objectives.

*2.3. Subjective Video Quality Assessment*

In order to provide the best entertainment and marketing experience, some businesses have begun investigating whether there is a relationship between video quality and consumer satisfaction. Messai et al. [33] emphasized that image quality assessment (IQA) plays an essential role in diverse image-related processes and applications. There are two types of quality evaluations, subjective methods based on human opinion scores and objective methods based on parameters or computational models. Varga and Szirányi [34] contend that subjective evaluations offer a more trustworthy and precise method for assessing digital video quality. In order to obtain more rigorous evaluation results, a subjective evaluation should be conducted in conjunction with established recommendations or literature. For instance, ITU-R. [35] outlined specific experimental protocols, equipment prerequisites, and analytical techniques tailored for SVQA measurements. These included what type of observation mode (with or without reference images) was required for subjects, how the experimental environment was constructed, and how to collect the data. Varga [36] simplifies this by stating that the core approach of SVQA involves gathering users' subjective opinions on each video's quality and subsequently deriving an average opinion score. SVQA is a useful tool for assessing the impact of parameter changes on user experience, since a cost-effective image enhancement effect will be more valuable if the user perceives a significant change in image quality [37]. The actual application environment for video is not limited to the laboratory, and the subtle objective differences between different quality

videos may become meaningless when they are confronted with the effects of people, environment, or hardware. Zhou et al. [38] underscored that, historically, SVQA has been a dependable metric for justly evaluating image quality. The challenge with SVQA is ensuring observational rigor while controlling costs, a concern that computer vision-driven video quality assessments also grapple with. Despite this, some studies, developing an automated video quality evaluation, attempt to make their objective evaluation results consistent with the subjective evaluation results of human scoring so as to demonstrate the model's robustness [39]. This study aims to evaluate consumers' SVQA responses to mobile interactive video ads and to ascertain if variations in the interactive narrative influence their video quality judgments.

### 2.4. Immersion

Morélot et al. [40] define immersion as the ability of a digital system to simulate a lifelike experience, encompassing attributes such as inclusiveness, extensiveness, surroundness, and vividness. By its nature, a heightened immersion level inherently suggests an enhanced interactive experience, filled with positive emotions. According to previous studies, the level of immersion is the objective ability of the system to provide sensory stimulation to the user through the use of technology [41]. Such immersion encompasses visual and auditory sensations, and in certain systems, it might even extend to tactile and olfactory senses. Daassi and Debbabi [42] highlight that users can engage with immersive digital content that mirrors the multisensory experiences of the real world. To put it another way, while the user interacts with the system, his or her audiovisual system can simultaneously receive information and stimulation. Tang et al. [43] observe that with the advent of immersive technologies, the lines separating the real from the virtual are increasingly becoming indistinct. Due to this, some classic explanations for immersion include the difficulty of feeling the passage of time by the user. Tang et al. [44] point out that while immersion and flow share similarities, flow represents an all-or-nothing state of presence, while immersion possesses nuanced layers. Alternatively, immersion can be classified into several distinct levels based on its intensity. Comparatively, flow experiences emphasize the balance between challenge and ability in contrast to immersion experiences [45]. Due to the fact that viewers are not faced with urgent tasks while watching interactive videos, immersion is chosen as the measurement variable. Shin [46] notes that immersion, a symbiosis between the system and user, can enhance a system's efficacy. Hsieh et al. [47] state that achieving immersion often entails eliciting arousal and delight in the user's interactive journey. Considering the importance of immersion in measuring the design effects of interactive systems, we have incorporated this concept into our research framework.

### 2.5. Satisfaction

Bakti et al. [48] argue that satisfaction encompasses more than just the evaluation of a product or service's inherent attributes, it also extends to external considerations like image, cost, and other aspects. Individuals may determine their overall satisfaction based on the results of their use or experience. Hepola et al. [49] describe satisfaction within the confirmation–disconfirmation paradigm as the outcome of juxtaposing prior expectations with actual encounters. The underlying idea is that consumers exhibit higher satisfaction when their actual experiences surpass notably modest expectations. Rodríguez et al. [50] assert that users base their evaluations of services on past experiences. Due to this, enterprises should not only pay attention to the products and services they provide, but also consider the impact that expectations have on customer satisfaction. Besides consumers' own experiences, expectations may also be influenced by other consumers' descriptions of their experiences. A high level of satisfaction among existing consumers may generate an important incentive for other consumers to engage in this type of shopping. Melović et al. [51] believe that transferring such positive experiences can notably bolster the potency of digital marketing strategies. While positive promotional impacts can emerge over time, they could potentially oscillate toward a less favorable outlook eventually. While much of the current

satisfaction research spotlights transient emotions, Li et al. [52] emphasize the evolving nature of consumer contentment over time. In order to measure consumer satisfaction with mobile interactive advertisements, we have included this variable in the survey. Measuring satisfaction allows us to gain a better understanding of the overall evaluation results of consumers when adjusting interactive narrative design methods.

*2.6. Purchase Intention*

Lu and Chen [53] assert that purchase intention directly underpins and determines actual buying behavior. Market researchers often consider improving consumers' purchase intentions as one of their most important objectives. In recent years, businesses have increasingly acknowledged the role of human–computer interaction systems in shaping consumer purchase decisions. Jung et al. [54] found that interactive environments boost consumers' willingness to buy products. Because the virtual environment provides consumers with an interactive experience, they may obtain product-related information. Yu and Huang [55] posit that engaging in game-like experiences can evoke psychological reactions in consumers, potentially driving actions like purchases. Mobile interactive video advertisements must pay attention to consumers' psychological reactions, as each node may require them to perform activities similar to those required for games. Diwanji and Cortese [56] highlight that user-generated videos tend to be more effective in prompting purchases than brand-generated ones. Unlike traditional video ads, mobile interactive video ads necessitate collaborative creation by both consumers and brands. Therefore, mobile interactive advertising may have a different impact on consumers' purchase intentions than traditional video advertising. It has been demonstrated in previous research that interactive technology, particularly interactive video, can be helpful when making a purchase decision. Rausch and Kopplin [57], for instance, point out that consumers seek extensive product information before deciding to buy. Technologies like panoramic and close-up images can help fulfill this informational need. Meng et al. [58] emphasize that the product type can impact ad design effectiveness, with different product ads conveying positive emotions influencing consumer decisions variably. We therefore divided product types according to product involvement and evaluated the impact of mobile interactive video advertising's interactive narrative on consumers' purchase intentions based on this premise.

**3. Research Method**

*3.1. Sample Identification*

Our approach breaks down the creation of interactive video advertisements into four key dimensions: text structure, interaction form, video content, and product involvement. We derived and categorized these dimensions from commercial videos currently circulating in the market. Using this classification, we produced diverse interactive video advertisements as research samples, ensuring controlled variables. Recognizing the potential influence of video quality on consumer perceptions, we standardized all sample parameters to match commercial standards. These parameters include image size, frame rate, color representation, encoding format, dynamic range, both constant and variable bit rates, scanning techniques (interlaced vs. progressive), and chroma sampling.

3.1.1. Selection of Text Structure and Interaction Form

Gu, Lin, Sun, Yang, Chen, Jiang, Miao and Wei [37] recommend classifying interactive narrative video advertisements by their text structure and interactive characteristics. The literature identifies five distinct text architecture categories. From these, five specific architectures are selected as representative examples, namely, the flowchart, adjusted, sea anemone, pure mesh, and track-switching structures. Additionally, the study identifies three interaction forms, which are single click, shake, and sliding. Figures 1 and 2 showcase the classification and selection outcomes for both text structure and interaction form.

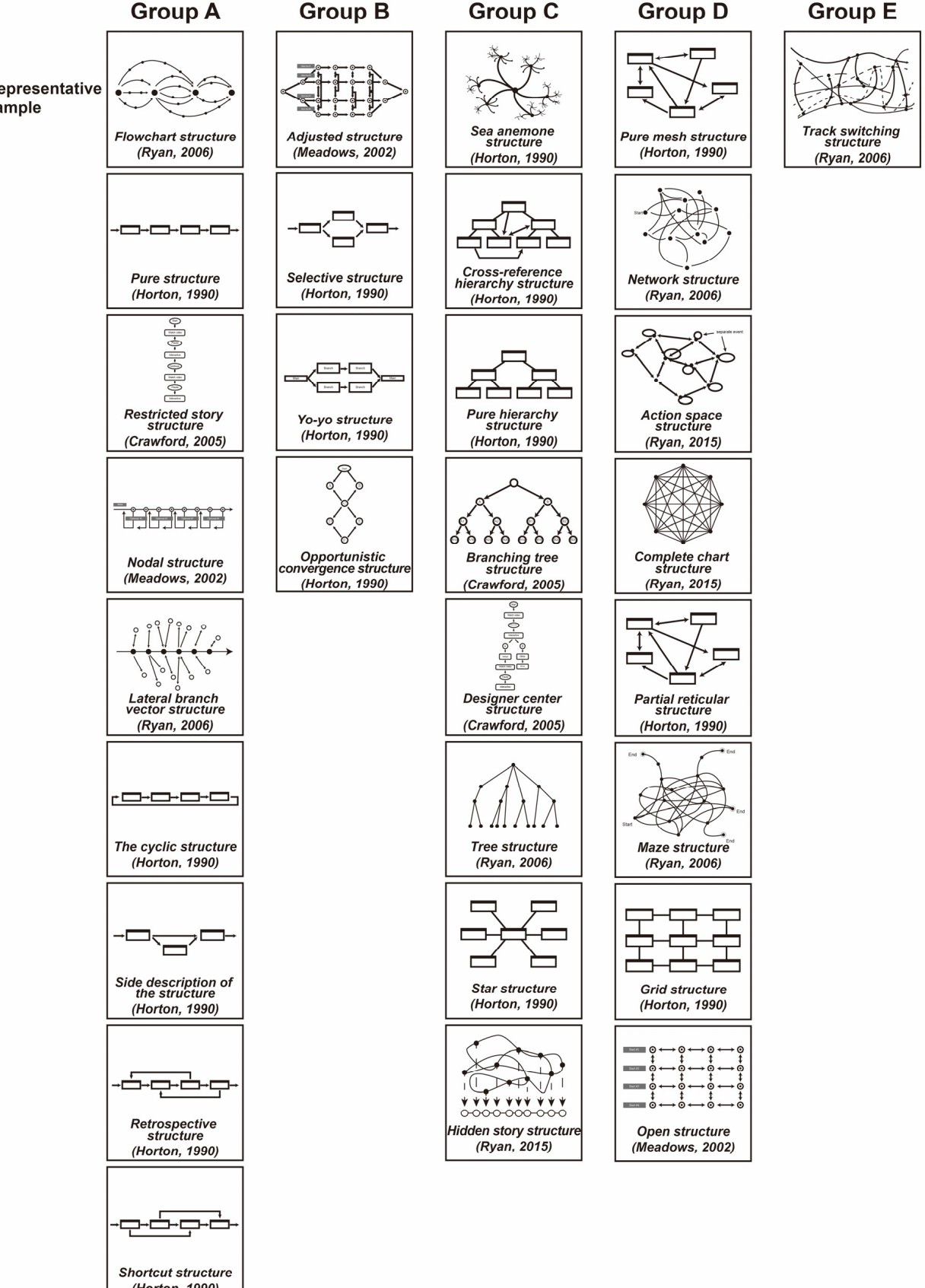

**Figure 1.** Text structure grouping and selection [12,59–62].

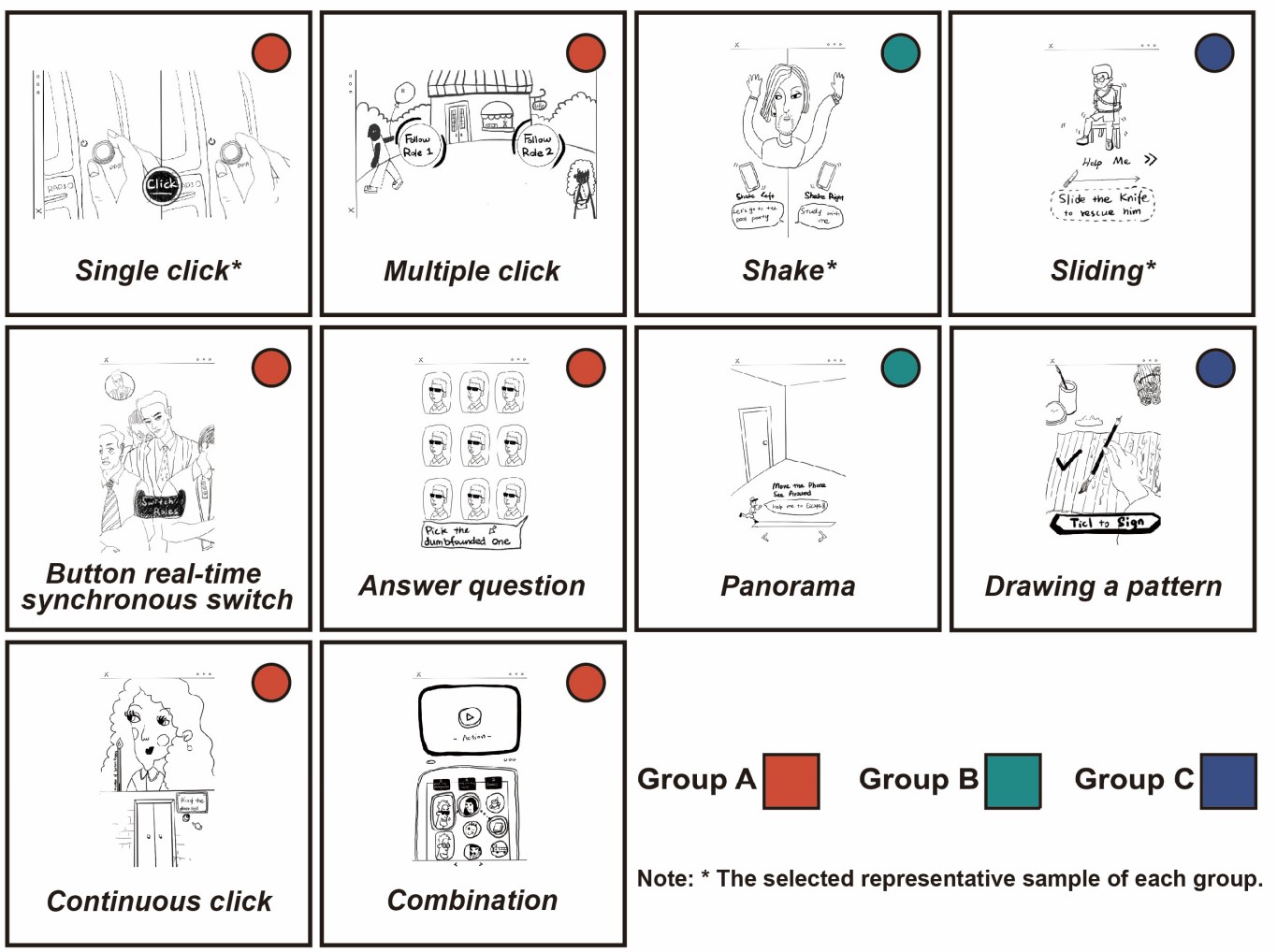

**Figure 2.** Interaction form grouping and selection.

3.1.2. Selection of Video Content

We collected 40 interactive video advertisements for this study, which included titles such as "Marketers have a history" and "Wangyou Town". During this research phase, we aimed to identify a narrative content template to use as a reference for crafting our interactive video samples. Thus, the design attributes and objective parameters of the interactive narratives in the gathered videos might influence expert evaluations. We stripped all interactive elements from the videos, converting them into a linear format. To standardize, we adjusted the video quality to 480 × 852 resolution, with 24 frames per second, 8-bit depth, and H264 encoding in an MP4 format. The video was set to a constant bitrate, employed progressive scanning, and used a 4:2:0 chroma sampling. Five experts were invited to a focus group session, where they were tasked with selecting videos they deemed to have outstanding content. The criteria for expert selection were based on standards used to evaluate renowned design industry awards, considering aspects like idea, form, function, differentiation, and impact. Details about the experts can be found in Table 1.

**Table 1.** Focus group expert information.

| Participants | Seniority | Position | Specialty and Work Items |
|---|---|---|---|
| ExpertsA | 5 years | Producer | Video production |
| ExpertsB | 10 years | Interaction designer | Interaction design |
| ExpertsC | 20 years | Assistant professor | Media art |
| ExpertsD | 10 years | Lecturer | Animation creation |
| ExpertsE | 10 years | Lecturer | Interaction design |

3.1.3. Selection of Product Involvement

An advertisement's product involvement can potentially influence consumer behavior. Our survey was tailored to focus on the 27 primary products of mobile interactive video advertisements as outlined by Gu, Lin, Sun, Yang, Chen, Jiang, Miao and Wei [37], along with their 8 corresponding classification outcomes. Through analysis of respondents' assessments of the representative products from each category, we identified products with both high and low degrees of involvement.

*3.2. Survey Procedure*

Interviews from prior studies with experts suggest that the primary demographic for mobile interactive video advertisements in Taiwan is young individuals aged 15 to 29 [37]. Consequently, we enlisted 60 and 100 young participants, organizing two questionnaire surveys in northern Taiwan with a strict gender-based division. We paid each subject TWD 400. Table 2 shows the subjects' details.

**Table 2.** Demographic characteristics of the respondents.

| Sample | Category | Survey 1 | | Survey 2 | |
|---|---|---|---|---|---|
| | | Number | Percentage (%) | Number | Percentage (%) |
| Gender | Male | 30 | 50.000 | 50 | 50.000 |
| | Female | 30 | 50.000 | 50 | 50.000 |
| Age | 15–18 | 10 | 16.667 | 22 | 22.000 |
| | 18–22 | 29 | 48.333 | 43 | 43.000 |
| | 23–25 | 21 | 35.000 | 35 | 35.000 |
| Experience | Yes | 40 | 66.667 | 64 | 64.000 |
| | No | 20 | 33.333 | 36 | 36.000 |

The purpose of Survey 1 was to pinpoint products with high and low involvement in the developmental phase. Online surveys were completed by respondents in Survey 1. Each participant received a digital questionnaire, wherein they were prompted to evaluate previously analyzed representative products.

The purpose of Survey 2 was to assess the cumulative impact of text structure and interaction method. It is required that respondents in Survey 2 watch all mobile interactive advertisement samples one by one in random order on the survey site that we have created. The core of mobile interactive video advertisements lies in providing a succinct promotional experience. In the current market, many mobile interactive video ads are designed to have an experience duration of 2 to 3 min. The total duration of our samples, covering all branches, aligns with the content template we referenced (Video 10), which is within 3 min. Given that participants would engage with the interactive samples multiple times and fill out surveys, we designed each sample to last about one minute to keep the research process efficient. While viewing the interactive video advertisement, subjects were allowed to interact with it in a way that suited their preferences. In this framework, subjects can operate according to the interaction form provided by the advertisement and determine

the direction in which the story is directed. Before viewing the next sample, subjects were asked to complete a questionnaire regarding the current sample.

### 3.3. Questionnaire Design

In this study, the questionnaire we employed was constructed around mature items validated in earlier research, utilizing a 5-point Likert scale. The items for Survey 1 were sourced from the Bauer et al. [63] set of 12 questions. For Survey 2, we adapted and verified 15 items from Gu, Lin, Sun, Yang, Chen, Jiang, Miao and Wei [37], encompassing 3 questions on interactive narrative, 3 on subjective video quality assessments, 3 on immersion, 3 on satisfaction, and 3 related to purchase intentions. After refining our questionnaire, we forwarded it to three domain experts for preliminary review. Their feedback helped ascertain that both the phrasing and the inquiry approach were apt. The specifics of the questionnaire can be found in Table 3.

**Table 3.** Content of the questionnaire.

| Constructs | Coding | Items | Stage |
|---|---|---|---|
| Product in-volvement | PI1 | This product tells other people sth. about me. | Survey 1 |
| | PI2 | It helps me express my personality. | |
| | PI3 | It does not reflect my personality * | |
| | PI4 | It is part of my self-image. | |
| | PI5 | It is not relevant to me * | |
| | PI6 | It does not matter to me * | |
| | PI7 | It is of no concern to me * | |
| | PI8 | It is important to me. | |
| | PI9 | This product is fun. | |
| | PI10 | I find it fascinating. | |
| | PI11 | I find it exciting. | |
| | PI12 | I am interested in it. | |
| Interactive narrative | IN1 | I think the interaction of the AD is meaningful | Survey 2 |
| | IN2 | I think the AD gave me a new perspective | |
| | IN3 | I think the interactive experience provided by this AD is diverse | |
| Subjective video quality assessment | SVQA1 | I think the design of this advertisement is exquisite | |
| | SVQA2 | I think the advertisement is aesthetically pleasing | |
| | SVQA3 | I think the colors in this AD are nice | |
| Immersion | IM1 | I'll be interested to see how this advertising thing goes | |
| | IM2 | I'm really focused on watching this AD | |
| | IM3 | I think watching this AD is experiencing something rather than just doing something | |
| Satisfaction | SA1 | I think this AD has an appropriate advertising length | |
| | SA2 | I think the advertisement is creative | |
| | SA3 | I think this AD gives a good user experience | |
| Purchase intention | PI1 | If I have a need, I'm more likely to buy the advertised product | |
| | PI2 | I will buy the product in the advertisement | |
| | PI3 | I will probably buy the advertised product | |

* Item is reverse scored.

*3.4. Site for Survey*

In line with the recommendations from the [35], our survey site's facilities and lighting adhered to the prescribed standards. The chosen indoor light source was the low-lighting D65. During the interaction process, the maximum observation angle relative to the normal remained below 30 degrees. Figures A1 and A2 in Appendix A provide detailed data results of our measurements.

- The screen's average brightness displaying only black levels was 0.98 LUX.
- In a dark room, the average brightness of a screen showing only white levels was 30.63 LUX.
- In such a setting, the brightness ratio of black-display to white-display was approximately 0.03.
- In our experimental lighting condition, an inactivated picture tube registered a screen brightness of 1.8 LUX.
- The peak brightness under these conditions reached 311.00 LUX.
- For an inactivated kinescope, the screen luminance's ratio to its peak was roughly 0.01.
- The background brightness directly behind the image monitor was measured at 22.70 LUX, while the peak brightness was 141.80 LUX. This means the background luminance was 0.16 times the peak of the image brightness.

## 4. Results

*4.1. Results of Video Content Selection*

In the first round of selection, we asked each expert to choose five videos they believed had the most valuable content. From their choices, seven videos emerged as top selections, each being picked at least twice. We then proceeded to the second round, where these seven videos were showcased again, and experts were tasked to shortlist three of them. By the end of this phase, Video 10 stood out, having been chosen by experts four times for its superior content. Hence, guided by expert consensus, we selected Video 10 as the foundational model for our subsequent video productions. Detailed results of this expert video selection process can be found in Table 4.

**Table 4.** Results of the video selection for excellent content.

| Rounds | Experts A | Experts B | Experts C | Experts D | Experts E |
|---|---|---|---|---|---|
| Round 1 (total frequency) | Video 6 (2 times) * | Video 2 (2 times) * | Video 2 (2 times) * | Video 3 (2 times) * | Video 8 (1 times) |
| | Video 10 (2 times) * | Video 6 (2 times) * | Video 3 (2 times) * | Video 13 (1 times) | Video 9 (1 times) |
| | Video 15 (1 times) | Video 10 (2 times) * | Video 7 (1 times) | Video 23 (1 times) | Video 19 (1 times) |
| | Video 18 (1 times) | Video 25 (1 times) | Video 24 (1 times) | Video 35 (2 times) * | Video 32 (2 times) * |
| | Video 32 (2 times) * | Video 40 (2 times) * | Video 40 (2 times) * | Video 38 (1 times) | Video 35 (2 times) * |
| Round 2 (total frequency) | Video 6 (2 times) | Video 2 (3 times) | Video 2 (3 times) | Video 3 (2 times) | Video 2 (3 times) |
| | Video 10 (4 times) * | Video 10 (4 times) * | Video 3 (2 times) | Video 6 (2 times) | Video 10 (4 times) * |
| | Video 40 (3 times) | Video 40 (3 times) | Video 40 (3 times) | Video 10 (4 times) * | Video 35 (1 times) |

* Videos selected for the most part.

### 4.2. Results of the Product Involvement Survey

To determine the level of product involvement, we engaged 60 participants in Survey 1. Utilizing the Kruskal–Wallis test, we evaluated the degree of product involvement from the data obtained. Our findings showcased a significance level below 0.05. This highlights a marked difference in involvement between the highest-ranked product (mobile phone with a mean score of 4.213) and the lowest-ranked one (fatigue driving detection system with a mean score of 2.947). Such a distinction implies varying degrees of consumer interest in these two products. Consequently, for the purpose of this study, the mobile phone and fatigue driving detection system were chosen as representative products. Detailed outcomes of the product involvement categorization can be viewed in Table 5.

**Table 5.** Results of the product involvement division.

| Groups | Representative Sample | Other Samples | | | |
|---|---|---|---|---|---|
| 1 (Mean) | PC game (M = 3.454) | Music software | Mobile phone games | Mobile phone input method | |
| 2 (Mean) | Search engine App (M = 3.421) | Browser APP | Takeout platform APP | Video APP | Take a taxi APP |
| 3 (Mean) | Retail stores (M = 3.172) | Transportation bureau | Bank | | |
| 4 (Mean) | Drinks (M = 3.560) | Cleanser | Dress | Diamond | |
| 5 (Mean) | E-commerce platform (M = 3.619) | News website | Live platform | Fundraising platform | |
| 6 (Mean) | Mobile phone (M = 4.213) * | Car | Electrical appliances | | |
| 7 (Mean) | Artificial intelligence company (M = 3.300) | Advertising company | Game company | | |
| 8 (Mean) | Fatigue driving detection system (M = 2.947) * | | | | |

\* Highest and lowest product involvement products.

### 4.3. Results of Sample Making in Research

We recreated Video 10 through a copy shoot, maintaining its core details. However, the original product was replaced with two different products: a high-involvement mobile phone and a low-involvement fatigue driving detection system. All interactive nodes from Video 10 were retained, and using them, we developed an interactive video, varying its text structures and interactive forms. The content was roughly divided into five chapters for better ambient lighting assessment. Chapter 1 had four segments, while Chapters 2 to 4 had eight segments each, irrespective of being Version A or B. These segments represent the four key stages of narrative development, as depicted in a table. Chapter 5 again consisted of four segments. Interactive nodes were strategically placed between segments, structuring the storyline. These nodes and the narrative blueprint can be found in Table 6. A sketch of our interactive video is illustrated in Figure 3. This video closely mimics Video 10, replicating its scenes and effects. The sketched version was created as a measure against potential copyright complications arising from similarities with Video 10.

To assess the impact of various interactive narrative designs, we crafted 30 mobile interactive video ads, incorporating 5 text structures, 3 interaction methods, and 2 products, each with different levels of involvement. This yielded 15 sets, each highlighting either high or low product involvement. The mix of text structure and interaction form is detailed in Table 7.

**Table 6.** Plot outlines and interaction nodes.

| Chapter | A High Level of Product Involvement (Mobile Phone) | A Low Level of Product Involvement (Fatigue Driving Detection System) |
|---|---|---|
| 1 | External connection to the hotel | |
| | Main character meets supporting role | |
| | Entry the hotel | |
| | Check in at the hotel | |
| 2A (high level) 2B (low level) | Enter the elevator after booking the Mobile phone α theme room | Enter the elevator after booking the fatigue driving detection system α theme room |
| | Go to the Mobile phone α theme room | Go to the fatigue driving detection system α theme room |
| | Enter the Mobile Phone α theme room | Enter the fatigue driving detection system α theme room |
| | Take a selfie in the Mobile Phone α theme room | Take a selfie in the fatigue driving detection system α theme room |
| 3A (high level) 3B (low level) | Enter the elevator after booking the Mobile phone β theme room | Enter the elevator after booking the fatigue driving detection system β theme room |
| | Go to the Mobile phone β theme room | Go to the fatigue driving detection system β theme room |
| | Enter the Mobile Phone β theme room | Enter the fatigue driving detection system β theme room |
| | Take a selfie in the Mobile Phone β theme room | Take a selfie in the fatigue driving detection system β theme room |
| 4A (high level) 4B (low level) | Enter the elevator after booking the Mobile phone γ theme room | Enter the elevator after booking the fatigue driving detection system γ theme room |
| | Go to the Mobile phone γ theme room | Go to the fatigue driving detection system γ theme room |
| | Enter the Mobile Phone γ theme room | Enter the fatigue driving detection system γ theme room |
| | Take a selfie in the Mobile Phone γ theme room | Take a selfie in the fatigue driving detection system γ theme room |
| 5 | A nighttime view of the hotel roof | |
| | On the hotel rooftop, the main character selects a suitable location for a selfie | |
| | Both the main character and the supporting role are looking at the same location when taking a selfie | |
| | A selfie contest between the protagonist and supporting cast is taking place | |

**Table 7.** Interactive Narrative Results.

| Interaction Form | Text Structure | | | | |
|---|---|---|---|---|---|
| | **A** | **B** | **C** | **D** | **E** |
| A | Group 1 | Group 4 | Group 7 | Group 10 | Group 13 |
| B | Group 2 | Group 5 | Group 8 | Group 11 | Group 14 |
| C | Group 3 | Group 6 | Group 9 | Group 12 | Group 15 |

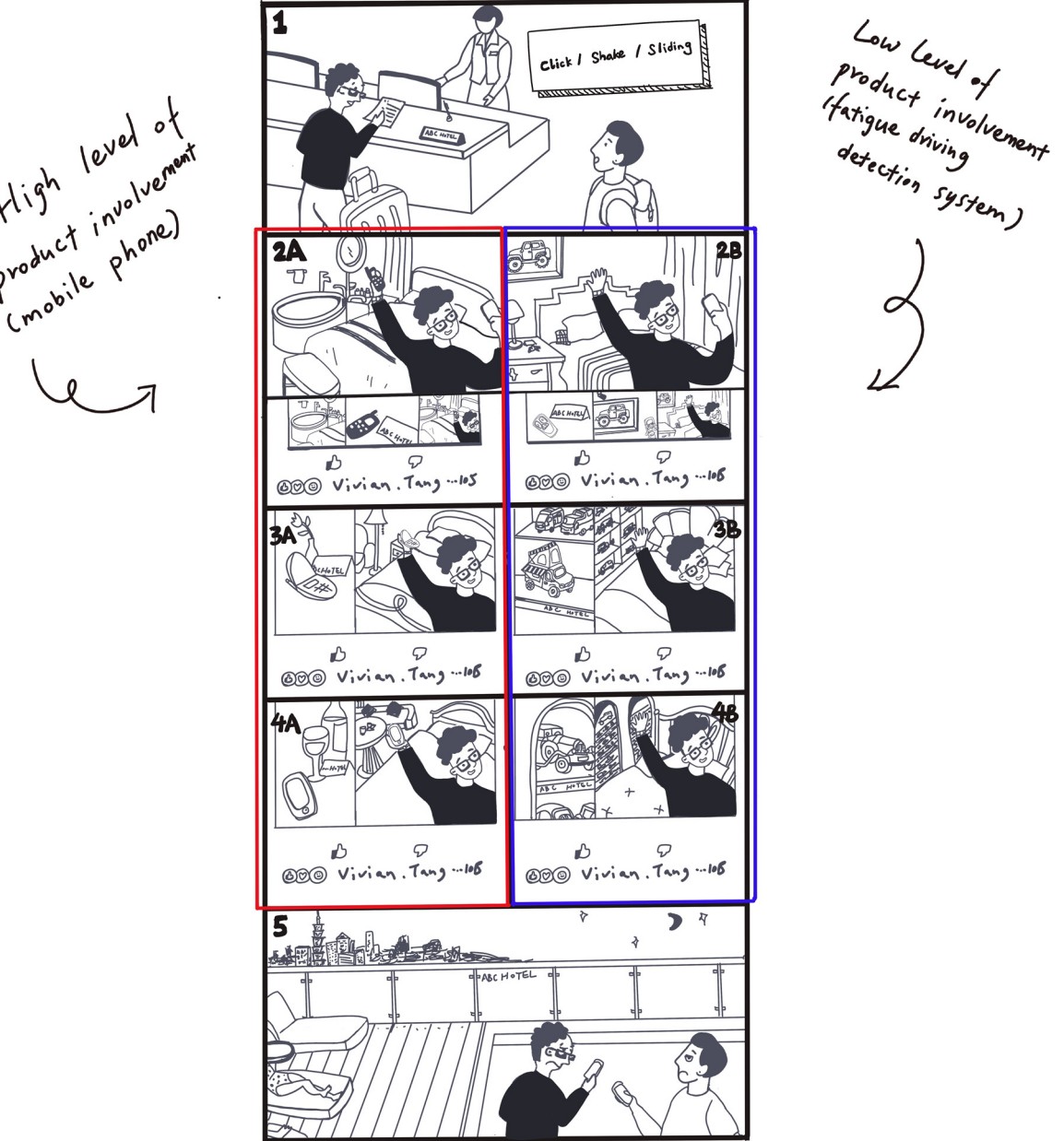

**Figure 3.** Video story.

*4.4. Combination of Interaction Form and Text Structure*

We invited 100 participants to participate in Survey 2, aimed at assessing the different effects resulting from various combinations of interaction forms and text structures. Each participant viewed the 30 randomly ordered samples we produced (15 highlighting high product involvement and 15 emphasizing low product involvement). While each sample was unique, covering content from Chapters 1 to 5 (including just one version, either A or B, for Chapters 2–4), the high-involvement samples mirrored their low-involvement counterparts, differing only in the promoted product. Thus, we garnered 3000 responses, yielding 100 evaluations per video sample. Using the collected data, we conducted two difference analyses through multivariate analysis of variance (MANOVA), differentiating between low and high product involvement.

We used the cross-loading method alongside the Fornell–Larcker criterion to ensure the data's reliability and validity. Table 8 demonstrates that the factor loadings for items on their designated constructs surpassed those on alternative constructs, affirming the absence

of cross-loading issues. Additionally, Table 9 shows that the correlation coefficients between variables were consistently below 0.8, alleviating concerns of multicollinearity. These correlation coefficients were also found to be lower than the square root of the associated average variation extraction (AVE), with all AVE values being above 0.5. Furthermore, the composite reliability (CR) metrics were all above 0.7. These results strengthen our assertion regarding the discriminant validity of the study's constructs [64,65].

**Table 8.** Results of Cross Loadings.

| Items | Construct | | | | |
|---|---|---|---|---|---|
| | IN | SVQA | IM | SA | PI |
| IN1 | **0.903** | 0.571 | 0.648 | 0.674 | 0.568 |
| IN2 | **0.906** | 0.564 | 0.653 | 0.684 | 0.545 |
| IN3 | **0.904** | 0.574 | 0.626 | 0.670 | 0.545 |
| SVQA1 | 0.564 | **0.930** | 0.575 | 0.639 | 0.552 |
| SVQA2 | 0.606 | **0.944** | 0.617 | 0.685 | 00.602 |
| SVQA3 | 0.595 | **0.930** | 0.638 | 0.697 | 0.604 |
| IM1 | 0.654 | 0.613 | **0.916** | 0.713 | 0.674 |
| IM2 | 0.658 | 0.620 | **0.929** | 0.702 | 0.678 |
| IM3 | 0.643 | 0.563 | **0.905** | 0.672 | 0.650 |
| SA1 | 0.605 | 0.636 | 0.659 | **0.876** | 0.641 |
| SA2 | 0.671 | 0.621 | 0.649 | **0.874** | 0.587 |
| SA3 | 0.710 | 0.660 | 0.707 | **0.907** | 0.661 |
| PI1 | 0.594 | 0.597 | 0.718 | 0.691 | **0.950** |
| PI2 | 0.572 | 0.599 | 0.673 | 0.664 | **0.954** |
| PI3 | 0.579 | 0.598 | 0.688 | 0.676 | **0.952** |

**Table 9.** Results of Fornell–Larcker criterion.

| Construct | AVE | CR | Correlation coefficient | | | | |
|---|---|---|---|---|---|---|---|
| | | | IN | SVQA | IM | SA | PI |
| IN | 0.752 | 0.901 | 0.867 | | | | |
| SVQA | 0.827 | 0.935 | 0.629 * | 0.909 | | | |
| IM | 0.773 | 0.911 | 0.710 * | 0.651 * | 0.879 | | |
| SA | 0.723 | 0.887 | 0.747 * | 0.720 * | 0.758 * | 0.850 | |
| PI | 0.877 | 0.955 | 0.611 * | 0.627 * | 0.727 * | 0.711 * | 0.936 |

* The level of significance is 0.05.

We employed the Box' M test to verify if the covariate matrix aligns with the prerequisites for MANOVA, and utilized Levene's test to ascertain the homogeneity of the variables. As per Table 10, the covariate matrices between the variables are consistent ($p > 0.05$), and there is no significant difference in variances ($p > 0.05$). Therefore, MANOVA is deemed appropriate for data analysis.

**Table 10.** Results of homogeneity and equality of covariance matrices.

| Product Involvement | Levene's Test | | | Box's Test | | |
| --- | --- | --- | --- | --- | --- | --- |
| | Construct | Leaven Statistic | Sig. | Box's M | F | Sig. |
| Low level | IN | 0.920 | 0.537 | 218.430 | 1.020 | 0.406 |
| | SVQA | 0.592 | 0.873 | | | |
| | IM | 1.328 | 0.183 | | | |
| | SA | 0.631 | 0.841 | | | |
| | PI | 0.485 | 0.942 | | | |
| High level | IN | 0.944 | 0.510 | 208.324 | 0.973 | 0.598 |
| | SVQA | 0.240 | 0.998 | | | |
| | IM | 0.354 | 0.986 | | | |
| | SA | 0.721 | 0.755 | | | |
| | PI | 0.562 | 0.896 | | | |

Table 11 presents the results of the difference analysis. For mobile interactive video ads with low product involvement, consumers' perception of interactive narrative, immersion, and satisfaction varies based on the type of interaction form and text structure. The mean difference reaches the significant criterion with a small effect size (partial $\eta_2 > 0.009$). Other aspects did not show significant differences. In contrast, for these ads involving a high degree of product involvement, different interaction forms and text structures lead to a small effect size (partial $\eta_2 > 0.009$). Other aspects did not show significant differences.

**Table 11.** Difference analysis results for high-level product involvement.

| Product Involvement | Construct | Type III Sum of Squares | df | Mean Square | F | Sig. | Partial Eta Squared |
| --- | --- | --- | --- | --- | --- | --- | --- |
| Low level | IN | 29.018 | 14 | 2.073 | 2.026 | 0.013 * | 0.019 |
| | SVQA | 3.892 | 14 | 0.278 | 0.314 | 0.992 | 0.003 |
| | IM | 60.132 | 14 | 4.295 | 3.825 | 0.000 * | 0.035 |
| | SA | 36.208 | 14 | 2.586 | 2.498 | 0.002 * | 0.023 |
| | PI | 11.683 | 14 | 0.834 | 0.732 | 0.743 | 0.007 |
| High level | IN | 27.842 | 14 | 1.989 | 1.846 | 0.028 * | 0.017 |
| | SVQA | 5.517 | 14 | 0.394 | 0.395 | 0.977 | 0.004 |
| | IM | 12.560 | 14 | 0.897 | 0.723 | 0.752 | 0.007 |
| | SA | 10.215 | 14 | 0.730 | 0.697 | 0.779 | 0.007 |
| | PI | 8.485 | 14 | 0.606 | 0.494 | 0.937 | 0.005 |

* The level of significance is 0.05.

Using the Duncan method, we compared the evaluation results for interactive narrative, immersion, and satisfaction in mobile phone interactive video ads with low product involvement. We identified homogeneous subsets within these evaluations, with mean and standard deviation (SD) presented in Table 12. For the interactive narrative, immersion, and satisfaction constructs, three overlapping homogeneous subsets were found within the group ($p > 0.05$). Within the interactive narratives category, 11 groups were identified in the top-rated cluster. For immersion, only Group 1 ranked among the top-rated clusters. Regarding satisfaction, 7 groups were identified as top-tier. Notably, Group 1 was the highest-rated across all three constructs. For the two constructs combined, the top-rated cluster included five groups: Group 2, Group 10, Group 11, Group 13, and Group 14.

**Table 12.** Results of low-level product involvement after testing.

| Indices | Group | IN | | | Group | IM | | | Group | SA | | |
|---|---|---|---|---|---|---|---|---|---|---|---|---|
| | | 1 | 2 | 3 | | 1 | 2 | 3 | | 1 | 2 | 3 |
| Mean (SD) | 9 | 2.989 (1.089) | | | 5 | 2.757 (1.017) | | | 4 | 2.767 (1.014) | | |
| | 4 | 3.000 (1.033) | | | 4 | 2.760 (1.039) | | | 6 | 2.800 (1.015) | | |
| | 7 | 3.013 (1.127) | | | 12 | 2.763 (1.150) | | | 12 | 2.813 (1.016) | | |
| | 3 | 3.033 (1.012) | 3.033 (1.012) | | 9 | 2.783 (1.108) | | | 5 | 2.820 (1.011) | | |
| | 6 | 3.040 (0.984) | 3.040 (0.984) | 3.040 (0.984) | 6 | 2.790 (1.060) | | | 8 | 2.863 (1.110) | 2.863 (1.110) | |
| | 5 | 3.050 (0.928) | 3.050 (0.928) | 3.050 (0.928) | 15 | 2.837 (1.124) | | | 9 | 2.863 (1.014) | 2.863 (1.014) | |
| | 12 | 3.067 (1.040) | 3.067 (1.040) | 3.067 (1.040) | 8 | 2.853 (1.098) | | | 15 | 2.923 (1.025) | 2.923 (1.025) | |
| | 15 | 3.120 (1.048) | 3.120 (1.048) | 3.120 (1.048) | 7 | 2.863 (1.156) | | | 7 | 2.927 (1.095) | 2.927 (1.095) | |
| | 8 | 3.137 (1.070) | 3.137 (1.070) | 3.137 (1.070) | 11 | 2.943 (1.120) | 2.943 (1.120) | | 11 | 3.023 (1.053) | 3.023 (1.053) | 3.023 (1.053) |
| | 2 | 3.167 (0.986) | 3.167 (0.986) | 3.167 (0.986) | 3 | 2.967 (0.984) | 2.967 (0.984) | | 13 | 3.037 (1.061) | 3.037 (1.061) | 3.037 (1.061) |
| | 1 | 3.290 (0.881) | 3.290 (0.881) | 3.290 (0.881) | 13 | 2.980 (1.067) | 2.980 (1.067) | | 3 | 3.040 (1.019) | 3.040 (1.019) | 3.040 (1.019) |
| | 14 | 3.293 (0.949) | 3.293 (0.949) | 3.293 (0.949) | 14 | 2.983 (1.037) | 2.983 (1.037) | | 10 | 3.107 (1.034) | 3.107 (1.034) | 3.107 (1.034) |
| | 10 | | 3.350 (1.052) | 3.350 (1.052) | 10 | 3.020 (1.081) | 3.020 (1.081) | | 14 | 3.107 (1.005) | 3.107 (1.005) | 3.107 (1.005) |
| | 13 | | | 3.370 (0.949) | 2 | | 3.220 (0.895) | | 2 | | 3.190 (0.885) | 3.190 (0.885) |
| | 11 | | | 3.373 (0.993) | 1 | | | 3.527 (0.918) | 1 | | | 3.317 (0.877) |
| | Sig. | 0.078 | 0.064 | 0.052 | Sig. | 0.157 | 0.108 | 1.000 | Sig. | 0.052 | 0.057 | 0.079 |

For mobile interactive video ads with high product involvement, we focused solely on comparing the evaluation results of interactive narratives, detailed in Table 13. Four overlapping homogeneous subgroups evaluated the interactive narrative ($p > 0.05$). The findings revealed that 12 groups were part of the highest-rated cluster. Given the presence of a cross-factor effect on post-test outcomes, we highlight two groups, Group 10 and Group 13, which are significantly distinct from the combinations in the two lowest-scoring groups.

**Table 13.** Results of high-level product involvement after testing.

| Indices | Group | IN | | | |
|---|---|---|---|---|---|
| | | 1 | 2 | 3 | 4 |
| Mean (SD) | 7 | 2.970 (1.124) | | | |
| | 9 | 3.040 (1.150) | 3.040 (1.150) | | |
| | 3 | 3.060 (1.049) | 3.060 (1.049) | 3.060 (1.049) | |
| | 8 | 3.103 (1.066) | 3.103 (1.066) | 3.103 (1.066) | 3.103 (1.066) |
| | 15 | 3.170 (1.078) | 3.170 (1.078) | 3.170 (1.078) | 3.170 (1.078) |
| | 6 | 3.190 (1.000) | 3.190 (1.000) | 3.190 (1.000) | 3.190 (1.000) |
| | 2 | 3.207 (0.998) | 3.207 (0.998) | 3.207 (0.998) | 3.207 (0.998) |

**Table 13.** *Cont.*

| Indices | Group | IN | | | |
|---|---|---|---|---|---|
| | | **1** | **2** | **3** | **4** |
| Mean (SD) | 1 | 3.240 (1.025) | 3.240 (1.025) | 3.240 (1.025) | 3.240 (1.025) |
| | 5 | 3.277 (0.978) | 3.277 (0.978) | 3.277 (0.978) | 3.277 (0.978) |
| | 12 | 3.283 (0.993) | 3.283 (0.993) | 3.283 (0.993) | 3.283 (0.993) |
| | 11 | 3.307 (1.037) | 3.307 (1.037) | 3.307 (1.037) | 3.307 (1.037) |
| | 4 | | 3.367 (0.928) | 3.367 (0.928) | 3.367 (0.928) |
| | 14 | | 3.380 (1.028) | 3.380 (1.028) | 3.380 (1.028) |
| | 13 | | | 3.397 (1.094) | 3.397 (1.094) |
| | 10 | | | | 3.440 (1.006) |
| | Sig. | 0.056 | 0.055 | 0.058 | 0.058 |

## 5. Discussion

To elevate consumers' interactive experience, our study delves into the role of interactive narratives in mobile interactive video advertisements, subsequently offering design recommendations rooted in our observations. Our findings suggest that mobile interactive video ads, when crafted using diverse interactive narrative techniques, can profoundly influence consumers' perceptions and preferences, underscoring the need for a structured design framework for interactive narratives. As highlighted by de Regt et al. [66], an apt interactive narrative design can amplify the consumer's engagement with interactive content—a sentiment echoed by our study. Beyond aiding businesses in meeting their marketing goals, our design insights can alleviate pressures on designers. By employing methodical and sound design strategies, designers' overreliance on personal experiences during the creation phase can be diminished. Consequently, our outcomes make the interaction design field more accessible, enabling designers to swiftly acquire and apply interaction design methodologies to produce precise and effective designs.

By merging diverse text structures with various interaction forms, we devised a consumer-centric interactive narrative design approach. Specifically, we categorized design strategies according to product involvement: those with high product involvement and those with low product involvement, providing tailored design recommendations for each. It is worth noting that our suggestions extend beyond the 27 product types assessed in this research. As noted by Bauer, Sauer and Becker [63], manufacturers can gauge the rapport between their products and their target audience using our adopted product involvement scale, comprised of a set of 12 questions.

Figure 4 outlines our suggested interactive narrative design strategies. Drawing from our research on consumer perceptions, we surmise that products categorized with low product involvement may benefit more from the six interactive narrative design methods displayed on the left side of the figure. Conversely, for products marked by high product involvement, the two combined methods on the figure's right side seem more apt. Detailed recommendations for these combinations and ensuing discussions are delineated below.

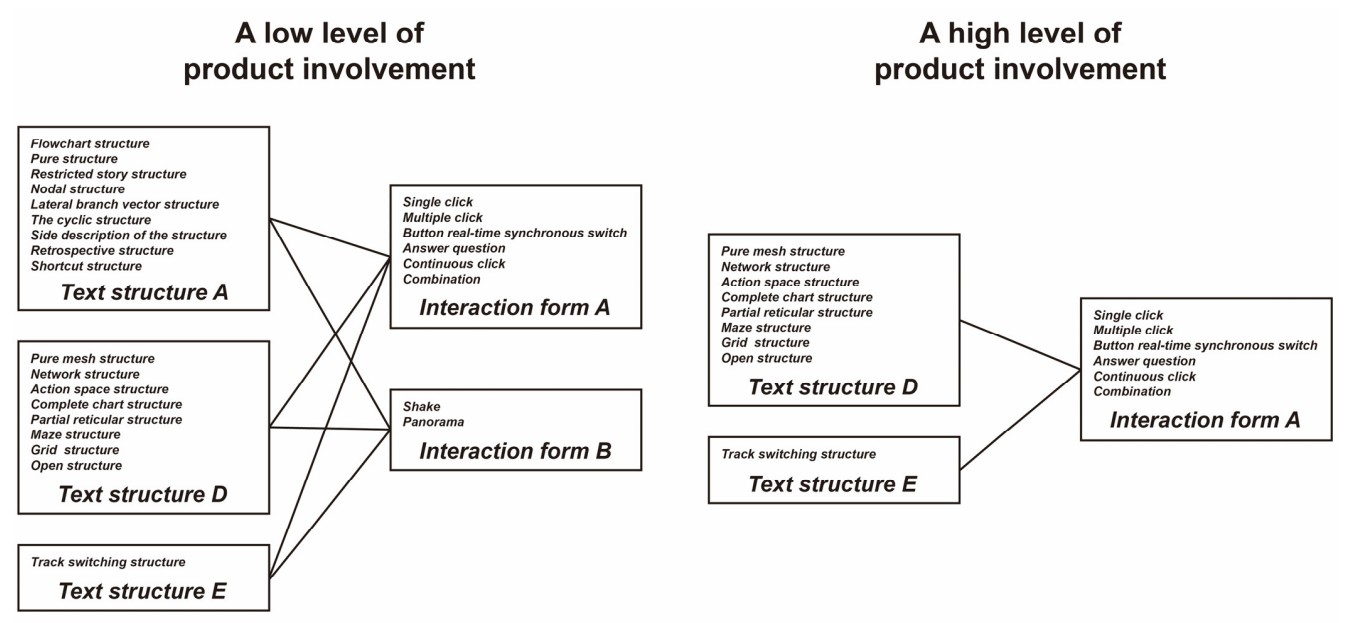

**Figure 4.** Recommended interactive narrative methods.

The results of our study show that consumers have different perceptions of the three constructs of video interactive narrative, immersion, and satisfaction when they are involved at a low level in the product development process. In terms of consumers' SVQA for videos and purchase intentions for products, there is no difference. We believe, however, that the relationship between interactive narrative design methods and marketing effectiveness cannot be ignored under this assumption. According to the results of Gu, Lin, Sun, Yang, Chen, Jiang, Miao and Wei [37], consumer interaction narrative design effects have a positive and fully mediated effect on purchase intention under the mediation of immersion and satisfaction. The study also shows that immersion and satisfaction can positively influence purchase intentions. Consumer purchases and immersion are strongly correlated, as noted in some marketing literature [67,68]. In our view, even if the combination of text structure and interaction form is not likely to have a significant impact on SVQA and purchase intention, it could still be combined with other conditions in actual marketing to influence consumers' decisions. Considering the importance of immersion and satisfaction in marketing campaigns, it is also important to choose the interactive narrative combination method correctly when marketing products with a low level of engagement.

To enhance consumer experiences via mobile interactive video advertisements, companies with low product involvement might consider adjusting the blend of text structure and interaction form to bolster design and marketing outcomes. We recommend that businesses and designers prioritize the six interactive narrative design techniques, namely, Group 1, Group 2, Group 10, Group 11, Group 13, and Group 14. Especially noteworthy is Group 1, which excels across all user perception metrics. This method employs flow charts as its text structure and utilizes click-based controls for interaction. While this approach seems elementary, it surprisingly yields optimal design outcomes. Within the realm of interaction design, showcasing advanced technological prowess with intricate user experiences might be tempting. However, complexity is not always the user's top preference. Oftentimes, straightforward and user-centric approaches pave the way for exemplary user experiences [69].

The other suggested combinations, similar to Combination 1, collectively offer a strong synergistic effect. Specifically, interactive types A and B, built upon click or gravity-sensing mechanisms, pair well with narrative structures A, D, and E. Research indicates that clicking, a timeless interaction method, is valuable for interactive narratives. Moreover, gravity sensing has been proven to enhance interactive behavior by affording users multiple device interaction options [70,71]. Contrasting other narrative structures, types A, D, and E

each have distinctive features. For one, narrative structure A prioritizes the director's vision; the storyline remains largely undisturbed by viewer interactions. Thus, videos with this narrative maintain the core plot envisioned by a director while enabling viewer interactivity. Secondly, narrative structure D offers a versatile plot progression. Its interconnected layout lets users freely explore, gravitating to ad segments they find captivating. Interactive videos leveraging this structure do not adhere strictly to a fixed storyline. Enthusiastic viewers can instantly jump to key scenes or results, and those keen on revisiting past segments can easily backtrack owing to the flexible structure. Thirdly, narrative structure E introduces an innovative narrative approach. It presents concurrent storylines, enabling audiences to journey through the narrative from multiple angles. They can select a character of interest and deeply engage with their narrative. As the plot advances and preferences change, viewers can fluidly shift their focus. Such combination approaches have garnered high appreciation among consumers for all mobile device interactive video ads. Our data indicate that these combinations might shape consumer assessments of interactive narratives, immersion, and overall satisfaction.

Notably, consumers rated mobile phone interactive video ads utilizing sliding as the primary interaction method quite low. Achieving an engaging narrative effect and immersion seems challenging with a sliding interface, irrespective of the text structure. We observed that type B, which reverts to the original plot post branching, and type C, tree-like in nature, fail to offer consumers a fulfilling interactive journey. In the case of type B, users seem to possess limited decision-making autonomy, leading them back to the initial plot regardless of their choices. Such restricted decision-making in type B could prevent users from influencing the plot's direction. This observation aligns with Elnahla [72] insights: interactive videos allowing users to craft their own narratives gain broader acceptance. While collaborative efforts and user interactivity enhance user experience, the opposite might not yield the anticipated results. Conversely, type C's potential shortcoming could stem from its overly branched storyline. User behavior studies reveal that the perceived simplicity of interactive tools directly boosts consumer satisfaction, enhancing their product perception [73]. Our research mirrors these findings, especially concerning user preferences for mobile interactive video ads.

Furthermore, with high product involvement, we find that the combination of text structure and interaction form will only result in differences in consumers' perception of interactive narrative constructs. Gu, Lin, Sun, Yang, Chen, Jiang, Miao and Wei [37] indicate that interactive narrative demonstrates an impact ranging from 0.688 to 0.856 on constructs like SVQA, immersion, satisfaction, and purchase intentions. Nevertheless, in our actual measurement, the evaluation results of these constructs did not significantly change as a result of the change in the interactive narrative approach. We conclude that while different interactive narrative combination methods have their pros and cons and have advantages and disadvantages, they cannot play a decisive role in determining consumer behavior when the product has high levels of engagement. Numerous factors influence marketing, including but not limited to, brand reputation and brand image [74], cleaner production processes and technologies [75], and subjective norms and self-control ability [76]. For high-involvement products, consumers might already be acquainted. The consumers have already set expectations for the product, and might even have compared it with other products on the market. At this time, consumers' past experiences and other factors are more likely to determine the effectiveness of advertising. Given that varying text structures and interactive narratives show significant differences in interactive narrative constructs, we believe that companies and designers cannot ignore interactive narratives when making mobile interactive video ads, especially for products with high involvement. Nevertheless, this design consideration must be integrated with other variables as part of the marketing management process. In the de Regt, Plangger and Barnes [66] study on interactive narratives, social influence and brand-related emotion are also relevant variables. Merely achieving excellent interactive narrative design might not guarantee the realization of marketing objectives. Based on the results of the study, Group 10 and Group 13 were

the only design methods that were significantly different from all of the combinations in the low-scoring group. The recommended interactive narrative solution combines mesh text structure or track-switching structure and click-based interaction. Interestingly, both methods perform well when the product involvement level is low. Employing strategies from either Group 10 or Group 13 proves to be a relatively safe and effective design approach for enterprises that are unable to accurately classify the product involvement of target consumers for objective reasons.

## 6. Conclusions

### 6.1. Contribution

This study builds upon previous research to examine and categorize methods of interactive narrative design specifically for mobile interactive video advertisements. Our findings suggest that interactive narratives, when structured in specific ways, can significantly influence marketing activities, especially in shaping consumer perceptions and preferences. Our contributions to this field are both theoretical and managerial, advancing the domain of interactive video marketing applications.

Our research results underscore the theoretical significance of interactive narratives in commercial advertising. Our findings not only lay a solid theoretical foundation but also directly validate the influence of interactive narrative design on consumer behavior. Additionally, we classified and measured the text structure and interaction form in this study based on prior classification recommendations. Evidence suggests that the practical classification and design process can substantially shape consumers' perceptions of interactive narratives, reinforcing the theoretical soundness of our classification approach.

From a managerial perspective, we offer design recommendations for mobile interactive video advertisements tailored for companies with both high and low product involvement. Previous studies have frequently analyzed variables influencing consumers' purchasing intent. Our research delves into the practical implementation of an effective design. To study the impact of interactive narratives' design, we employed experimental methods grounded in existing literature and with controlled variables. Enterprises and designers can swiftly choose an apt design method from our research findings, enabling them to craft an interactive marketing environment tailored to their requirements. Our research contributes to enhancing the user experience in human–computer interactions and bolsters the marketing efficacy of enterprises.

### 6.2. Limitations and Future Research

There are five main limitations to our research. Firstly, this study took place in Taiwan. The cultural background and local customs might influence consumer behavior and decisions. Future studies could sample from a variety of regions to ascertain if consumer experiences with interactive media remain consistent across different cultural contexts. Secondly, our survey only gauged consumer evaluations on five constructs: interactive narrative, SVQA, immersion, satisfaction, and purchase intention. There may, however, be other variables involved in consumers' decision-making during the marketing process. Future studies can expand on these variables, examining potential interaction effects between them and investigating the influence of interactive narrative design on yet-to-be-measured variables. Thirdly, our study primarily employs a quantitative approach. Certain nuanced aspects of user psychology and behavior might escape detection when solely relying on quantitative indicators. Consequently, grounded theory, action research, and other qualitative methods could be employed in future research to both validate our findings and delve deeper into consumers' perceptions of interactive narratives in mobile video ads. Fourthly, since our study simultaneously tested various sample types, we capped the video durations to respect participants' endurance and patience. Future inquiries might explore the ramifications of complete interactive video viewings on user experiences. Lastly and notably, both clients and interaction designers are predominantly concerned about the marketing efficacy of interactive advertisements as compared to traditional linear video

advertisements. We too are eager for a broader comparative study, especially one that dissects the impact of tech-driven interactive ads vs. traditional video ads on consumer purchasing decisions when brands are appropriately segmented and defined.

**Author Contributions:** Conceptualization, C.G.; methodology, S.L. and Y.Z.; software, W.W. and Y.Z.; validation, W.M. and Y.Z.; formal analysis, J.S.; investigation, J.C.; data curation, W.W. and W.M.; writing—original draft preparation, C.G. and J.S.; writing—review and editing, C.Y.; visualization, J.C.; supervision, S.L.; project administration, C.Y. and J.S. All authors have read and agreed to the published version of the manuscript.

**Funding:** This work was supported by Zhejiang A&F University [grant number 2023FR004].

**Data Availability Statement:** The data that support the findings of this study are available from the corresponding author upon reasonable request.

**Acknowledgments:** We are grateful to Fangfang Chen, Libing Wang, and Yinyin Zheng for their work in conceptualization. We also thank the anonymous reviewers who provided valuable comments on the manuscript.

**Conflicts of Interest:** The authors declare no conflict of interest.

## Appendix A

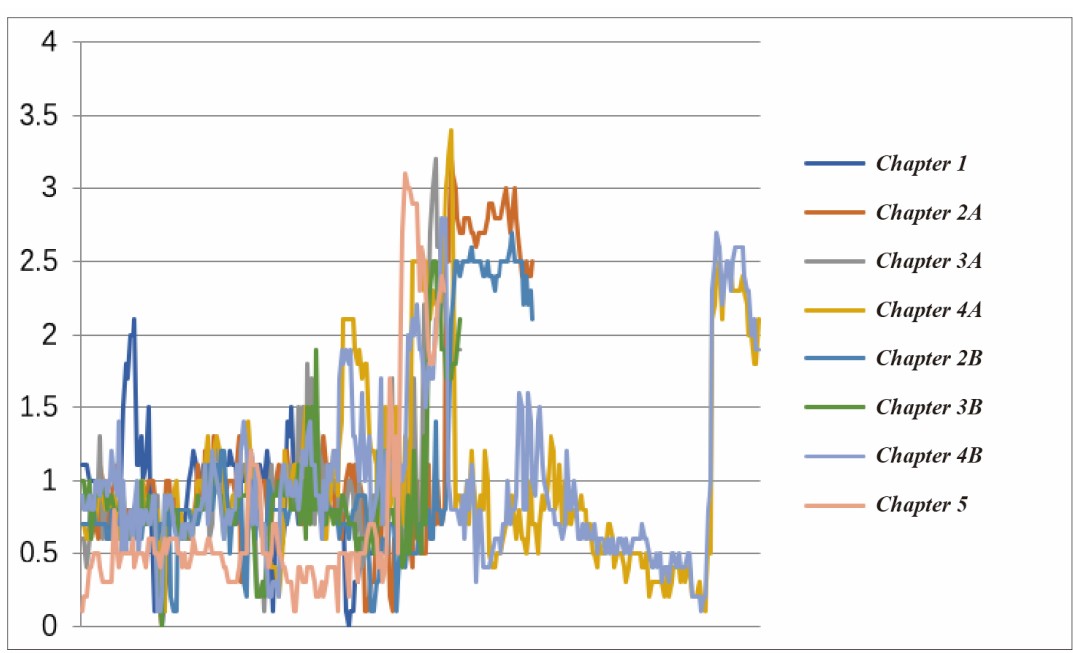

**Figure A1.** Graph of brightness fluctuations in a dark environment with only black levels displayed.

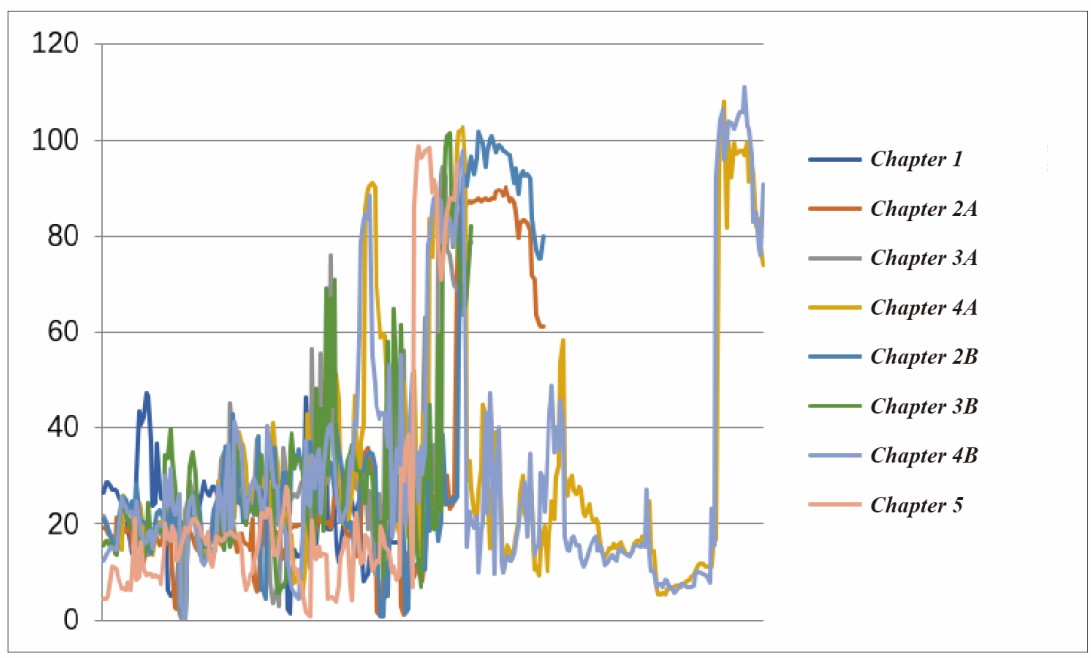

**Figure A2.** Graph of brightness fluctuation in a dark room when only the white level is displayed.

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
