# Peer review of "How Does Interactive Narrative Design Affect the Consumer Experience of Mobile Interactive Video Advertising?"

_systems, doi:10.3390/systems11090471_

Round 1
Reviewer 1 Report
I was very excited to read this paper, as I have over 20 years of experience conducting experiments with interactive design, including interactive narratives. However, I was very confused by the results. It appears there were 30 different interactive experiences available, 15 each for 2 different products. These were tested by a sample of 100 young people aged between 15 and 25. Does this mean that each person saw 30 experiences, in random order, flipping between the 2 products and the plot of the experience (going up an elevator to the rooftop of a hotel)? Or was the plot controlled, but experienced in 30 different ways (there are 40 chapters in Table 6)? Maybe each person did the experiment twice, once for the high-involvement product and again for the low-involvement product (ideally, product would be randomly ordered)? It is possible that I have misunderstood the experimental method and the results, because the writing was unclear. I hope the authors can present the results more clearly. Ideally, they would show example still pictures of the events and choices for each group, along with its sample size, mean, and standard deviation for each measure. The paper should report the correlations between the measures, and ideally, a factor analysis of the items to show they had no large cross-loadings (using maximum likelihood extraction and oblique rotation [Carpenter, 2018, https://doi.org/10.1080/19312458.2017.1396583]). No correlation should be larger than .80 (Rönkkö & Cho, 2022, https://doi.org/10.1177/1094428120968614).
More generally, before making claims that anyone should be using interactive ads, or different kinds of interactive narrative, the authors need to compare the results for interactive ads against those for professionally produced non-interactive ads for the same brands. Box office dollars show that linear movies do better than interactive experiences, because viewers value the decisions made by professional directors. But things may change as interactive technology (e.g., virtual reality) improves. Before ruling out the use of complex or simple branching narrative structures, versus simple linear interactive structures, the authors need to test experiences of the complete experience, not just a minute.
I wish the authors the best of luck with their future research.
The writing would benefit from the services of a paid, professional English editor.
Author Response
Dear reviewer,
We would like to express our sincere appreciation for your invaluable contributions to the peer-review process of our submission. Your insightful comments and constructive feedback have been instrumental in enhancing the quality of our manuscript.
We have carefully addressed all your suggestions and concerns in our response, which can be found in the attached document. Your expertise and guidance have been invaluable in refining our work, and we believe your feedback has significantly strengthened the scientific merit of our research.
Once again, we extend our gratitude for your dedication to the peer-review process, and we look forward to your continued support in shaping the final version of our paper.

Reviewer 2 Report
1. The research has applied user experience analysis, which considers all possible variables related to the topic by considering the user's initial issue. For example, it examines the impact of ads featuring the product on subsequent purchasing behavior and how ads influence consumer desire for the product.
2. Both literature review and discussion have provided a comprehensive analysis to integrate the current research with the past related research. Most of the issues included have been exposed in the literature review, which led the readers to follow the author's idea easily.
3. The research mentions perspectives on adopting interactive narratives as innovative methods to assist new storytelling possibilities. Besides, the authors also serve the perspective in both field (Industry) and academic applications to emphasize the potential of this topic in many domains.
4. The text classifications were well distinguished with the source that provide better analysis for the following steps of analysis applied. The objective video quality and requirements for the research experiment were well-prepared in detail to avoid bias from the different classifications or technical issues.
5. Multi-analyses showed systematic methods, from finding suitable videos to video development. The detailed procedure creates a systematic and limits the doubt of data development during the research.
6. Using mixed methods in the scientific approach provides more evidence to mitigate the subjective biases of the designer, one of the stakeholders involved in this research.
However, there are some suggestions as follows:
1. The site for the survey was a brilliant idea to include; however, if it does not make any significant results (or cannot show the effect of this effort on the survey result), involvement in the writing might not be necessary.
2. Figure 2 presents the interactive type proposed in this research. Yet, the group classifications are hard to follow, especially the order and the line distinguish visualization between groups A and C are pretty similar (suggest putting the number or changing the line).
3. In 3.1.2. The purpose of converting the video into the linear format would be better stated to let the readers understand the whole process and the necessity of implementing this step into the research.
4. The demography table shows a detailed distribution of the participants, yet only a few categories were used in this research. Suggesting that if the rest of the categories will not be put for further examination, it will be better to lay only the related demography into the table.
5. I don't get the chapter in Figures 3 & 4. Is there a relation to the chapters in Table 6? And if there was, please also explain why 8 chapters in Figures 3 & 4 while Table 6 has 38. Therefore, please clarify what "chapter" means here, as there is no further mention of this term besides the Figures and Tables.
6. Consider modifying one paragraph in the discussion, as it's pretty long. Itemizing or adding one table to explain the Group is encouraged here. Each result is partial, or they are supposed to be different from each other. The authors may consider further discussion regarding the relation between results or consider revising each part of the research, which was supposed to be one of the Five interactive narrative design methods.
7. Overall, some information is still needed for clarification, but considering the length of the paper, some parts should be removed or moved into the appendix. The missing Proof and showcases after all the collection of the chosen design elements questioning the validity of the research.
Author Response
Dear reviewer,
We wish to extend our heartfelt appreciation for your invaluable contributions as a reviewer for our submission. Your perceptive insights and constructive feedback have been pivotal in elevating the quality of our manuscript.
We are pleased to inform you that we have diligently addressed all of your insightful suggestions and concerns in our comprehensive response, which can be found enclosed in the accompanying document. Your expertise and guidance have been instrumental in refining our work, and we firmly believe that your feedback has greatly enhanced the scientific merit of our research.
Once again, we would like to express our gratitude for your unwavering dedication to the peer-review process, and we eagerly look forward to your continued support our paper.

Reviewer 3 Report
The article concerns mobile interactive video advertising. Authors propose interactive narrative design suggestions for mobile interactive video ads to enhance consumers perceptions, satisfaction and immersion.
The topic is very interesting and up-to-date in the modern world, because the interactive advertisements are more and more popular. This study examines and classifies the methods of interactive narrative design for mobile interactive video advertisements based on previous research.
I do not have any additional remarks concerning the methodology of research. In my opinion it is properly chosen and executed.
The results obtained in the study are interesting and the conclusion is consistent with the evidence and arguments presented. The main question is addressed. Authors properly identify the limitations of their study and present the directions for future research.
The references are appropriate - the provide solid background for the Authors' research. They are also up-to-date.
Tables and figures supplement the text of the paper in substantial way. I do not have any objections and recommendations concerning this element.
Some additional remarks:
1. Sometimes there no spaces between the reference and regular text of the paper (lines 38, 44, 160, ...)
2. I would like to ask Authors to provide access for the ads that they have created for the research, and give at least a link to a repository where they can be found.
Author Response
Dear reviewer,
We want to extend our sincere appreciation for your invaluable role as a reviewer for our submission. Your keen insights and constructive feedback have significantly enriched the quality of our manuscript.
We are pleased to inform you that we have diligently addressed all of your valuable suggestions and concerns in our detailed response, which can be found attached in the accompanying document. Your expertise and guidance have been instrumental in refining our work, and we firmly believe that your feedback has considerably strengthened the scientific merit of our research.
Once again, we would like to express our gratitude for your unwavering commitment to the peer-review process, and we eagerly anticipate your continued support in shaping the final version of our paper.
